# Review of Myeloma Therapies and Their Potential for Oral and Maxillofacial Side Effects

**DOI:** 10.3390/cancers13174479

**Published:** 2021-09-06

**Authors:** Sophie Beaumont, Simon Harrison, Jeremy Er

**Affiliations:** 1Peter MacCallum Cancer Centre, Melbourne, VIC 3000, Australia; 2Melbourne Dental School, University of Melbourne, Melbourne, VIC 3000, Australia; 3Sir Peter MacCallum Department of Oncology, University of Melbourne, Melbourne, VIC 3000, Australia; simon.harrison@petermac.org; 4Centre of Excellence for Cellular Immunotherapy, Peter MacCallum Cancer Centre, Melbourne, VIC 3000, Australia; 5Clinical Haematology, Peter MacCallum Cancer Centre and Royal Melbourne Hospital, Melbourne, VIC 3000, Australia; jeremy.er@petermac.org

**Keywords:** myeloma therapy, oral and maxillofacial side effects, medication related osteonecrosis of the jaw

## Abstract

**Simple Summary:**

Myeloma is a common cancer involving the bone marrow. Some of the medications used in the treatment of myeloma, including those that reduce the risk of bone fractures, can increase the chance of side effects occurring in the jawbone. The most serious complication in the jawbone is called medication-related osteonecrosis, meaning part of the jawbone dies. The aim of this review is to highlight some of the medications that are implicated and other risk factors that can contribute to osteonecrosis. Suggestions to prevent this complication from occurring are described. Conventional methods of treating established medication-related osteonecrosis of the jawbone are outlined as well as emerging new treatments.

**Abstract:**

Myeloma is a common haematological malignancy in which adverse skeletal related events are frequently seen. Over recent years, treatment for myeloma has evolved leading to improved survival. Antiresorptive therapy is an important adjunct therapy to reduce the risk of bone fractures and to improve the quality of life for myeloma patients; however, this has the potential for unwanted side effects in the oral cavity and maxillofacial region. Osteonecrosis of the jaw related to antiresorptive medications and other myeloma therapies is not uncommon. This review serves to highlight the risk of osteonecrosis of the jaw for myeloma patients, with some suggestions for prevention and management.

## 1. Introduction

Medication-related osteonecrosis of the jaw (MRONJ) is an uncommon but insidious side effect of some medications used in the treatment of multiple myeloma (MM). MRONJ may occur spontaneously after exposure to bone modifying agents such as bisphosphonates or denosumab or more commonly following an invasive procedure such as dental extractions. It can cause serious functional disturbance, morbidity, and adversely impact quality of life [1]. Although uncommon, MRONJ is in fact not a rare adverse event in myeloma patients; therefore, continuous monitoring and vigilance is essential from haemato-oncologists, and dental and oral maxillofacial specialists [2].

Treatment for MRONJ depends on its severity and clinical signs and symptoms. Preventing the onset of MRONJ is the most ideal goal as achieving complete resolution of MRONJ is often not possible [1]. Effective management relies on collaboration, patient education, and a multidisciplinary approach to patient care.

As the survival of MM patients improves with the discovery and development of novel therapies, the incidence of MRONJ may also rise as the duration of exposure to bone modifying agents increases. Hence, it is important to reduce the risk of developing MRONJ and optimise skeletal outcomes in patients with MRONJ. This review aims to summarise the mechanism of action of some of the common therapies used to treat myeloma and how these might impact the maxillofacial region.

## 2. Definition and Diagnosis of MRONJ

A diagnosis of MRONJ requires the following criteria: (1) current or prior treatment with a bone-modifying agent or angiogenic inhibitor; (2) exposed bone or bone that can be probed via a fistula through the mucous membrane or skin in the maxillofacial region; (3) persistent bone exposure of more than 8 weeks duration; (4) exclusion of metastatic disease or prior exposure of radiation therapy to the site [3].

## 3. Incidence

MM patients have a higher risk of MRONJ when compared with other patient cohorts taking antiresorptive medications, with the reported incidence ranging from 4.9–20.5% [4]. The incidence of MRONJ varies according to the type of medication, the duration of therapy, and the presence of other confounding factors, for example, increased age, comorbidities, and contributory medications [1,5,6,7]. The incidence also varies according to the underlying indication for antiresorptive therapy as summarised in Table 1 [8]. The incidence in patients on oral bisphosphonates, for example, clodronate, is lower when compared with zoledronic acid (1% vs. 4%, respectively) [9,10]. In one study of MM patients treated with bisphosphonates for a median of 19 months, MRONJ was diagnosed in 6.9% of patients, most of whom had a prior dental extraction due to dental/oral disease [11]. Another study of patients with advanced cancer treated with either zoledronic acid or denosumab, found 13/155 (8.4%) developed MRONJ, which was significantly associated with the number of infusions and the duration of antiresorptive therapy [12]. A recent literature review and meta-analysis including 13,857 patients reported the prevalence of MRONJ after receiving zoledronic acid for cancer ranged from 0.4–1.6% after one year, 0.8–2.1% after two years, and 1.3–2.3% after three years, with greater incidence after more years of administration [13]. ONJ incidence secondary to denosumab therapy ranged from 0.5% to 2.1% after one year, 1.1–3.0% after two years, and 1.3–3.2% after three years of treatment [13].

## 4. Pathogenesis

The potential mechanisms underlying the pathophysiology of MRONJ remain contentious with several hypotheses having been proposed (see Figure 1). Key mechanisms include inhibition of osteoclast differentiation and function, as these processes are integral to bone re-modelling and healing, and inflammation or infection associated with the dentition. The pathogenesis of MRONJ in myeloma patients is not unique. MRONJ primarily affects the alveolar jawbone of the maxilla and mandible. The alveolar processes of the jawbone have been shown to have the fastest bone metabolism in the skeleton [14], rendering these sites more susceptible to deposition of bisphosphonates [1]. The jawbone is also unique in that a close relationship exists between the teeth and the bone, facilitating a portal for microorganisms and other inflammatory agents to enter the bone as bacteria can stimulate bone resorption and contribute to bone necrosis [15]. Bisphosphonates can also inhibit the proliferation and viability of oral keratinocytes, damaging the oral mucosa and increasing the risk of infection [16]. Other proposed mechanisms include constant microtrauma in the jawbone resulting from occlusal forces from the dentition, inhibition of angiogenesis observed with zoledronic acid and anti-angiogenic agents, and suppression of innate or acquired immunity [1].

## 5. Risk Factors for MRONJ

### 5.1. Medications

More than 80% of patients with MM display evidence of myeloma bone disease, characterised by the formation of osteolytic lesions throughout the axial and appendicular skeleton [17] with an increased risk of fracture [18,19]. Treatment with bone modifying agents such as bisphosphonates and denosumab not only delay the time to the onset of skeletal related events (SRE) [20] but can improve bone pain in patients with cancer and bony metastases [21]. However, exposure to these drugs is a major risk factor for the development of MRONJ. Recent data suggest that combination therapies might also increase the risk of developing MRONJ and cause more advanced necrosis especially in the maxilla; however, this study was not limited to patients with myeloma [22]. Another study of 459 MRONJ cases reported that out of 52 patients undergoing treatment with BMA, 11 had also received lenalidomide, 12 received thalidomide, 11 received bevacizumab, 9 received everolimus, and 9 received sunitinib as part of the drug therapy [2]. Although bisphosphonates are the drugs most frequently associated with MRONJ, there is a growing range of non-antiresorptive medications implicated in MRONJ development [23].

#### 5.1.1. Bisphosphonates (BP)

Zoledronic acid and pamidronate are initial first-line therapy for management of myeloma bone disease as it inhibits bone resorption via actions on osteoclasts or osteoclast precursors [9]. Bisphosphonates bind to bone hydroxyapatite and inhibit osteoclast development, migration, and action, leading to decreased bone resorption without affecting the actual bone mineralisation. The concentration of BP is maintained at high levels within the bone lacunae for extended periods, leading to long-term osteoclast inhibition [11]. Non-nitrogen BP, such as clodronate, induces the apoptosis of osteoclasts via the accumulation of non-hydrolyzable ATP analogues. The more potent nitrogen-containing BPs, such as zoledronic acid and pamidronate, bind to hydroxyapatite crystals of bone and once incorporated into the osteoclast, inhibit the enzyme farnesyl pyrophosphate synthase blocking the mevalonate-pathway, leading to the accumulation of isopentyl diphosphate, the production of toxic metabolites leading to osteoclast apoptosis [9,24]. The Medical Research Council Myeloma IX trial showed that zoledronic acid was more efficacious in decreasing skeletal related events than non-nitrogen bisphosphonates and had a significant reduction in tumour burden in a subset of patients without skeletal fractures at presentation [10]. Zoledronic acid is also more favourable compared to pamidronate due to a reduced infusion time and less adverse events compared to pamidronate [9]; however, pamidronate can be used in patients with significant renal impairment.

In addition to its anti-osteoclastic effects, bisphosphonates also suppress angiogenesis through a direct inhibitory effect on the endothelial cells and can cause MRONJ by reducing blood vessel formation and impairment of post-interventional healing [25,26,27,28].

#### 5.1.2. Monoclonal Antibodies: Denosumab

Denosumab was first approved by the US Food and Drug Administration in 2010 for the treatment of osteoporosis in post-menopausal women [24]. Denosumab has a different mode of action to bisphosphonates. Denosumab is an anti-RANKL (receptor activator of nuclear factor kappa B ligand) monoclonal antibody that prevents osteoclastogenesis and osteoclast function by blocking the RANK–RANKL interaction [9]. This blockade inhibits key steps in osteoclast-mediated bone resorption [29]. Denosumab is metabolized and not stored in the bone and has a shorter half-life than bisphosphonates and is recommended when bisphosphonates are contraindicated in renal failure. It can also be given subcutaneously, allowing easier deliverability to patients. Denosumab has been demonstrated to be non-inferior to zoledronic acid for time to the first skeletal related events in a phase III randomized trial [30], with no difference in the incidence of ONJ between the two groups (4% vs. 3%) with a progression free survival benefit seen with denosumab in the autologous stem cell transplant-intent subgroup who received proteasome inhibitor based frontline regimens [31].

An overall resolution rate of ONJ of 36% was observed in an integrated analysis of three phase III trials of patients with metastatic bone disease receiving antiresorptive therapies, with a higher rate seen in denosumab (40.4%) compared to zoledronic acid (29.7%) [32]. This difference may be accounted for by the different mechanism of action of these agents. In contrast to the reversible inhibition of osteoclast by denosumab [33], there is accumulation of BPs within the bone and as it is released with bone resorption, it may cause further osteoclast inhibition [34].

Although generally safe, the risk–benefit effect on the long-term usage of bisphosphates and denosumab in the management of myeloma bony disease should be considered in order to to minimise bisphosphonate-associated adverse events. Another late effect of bisphosphonates is the development of atypical femoral fractures [35]. Although the pathogenesis is not fully understood, it is hypothesized to be related to long-term suppression of bone re-modelling leading to the accumulation of microdamage [36,37]. Bisphosphonates are adminstered up to two years from diagnosis of MM; however, risk stratification of SRE can be performed to adjust the scheduling of ongoing bisphosphonate therapy as suggested by Dickinson et al. and summarised in Figure 2 [38].

#### 5.1.3. Immunomodulators (IMiDs): Thalidomide, Lenalidomide, Pomalidomide

Following the discovery that thalidomide caused the suppression of tumour necrosis factor α (TNFα) and had powerful anti-angiogenic effects, it was reconsidered for use in cancer therapy [39]. Thalidomide was tested clinically in MM patients and demonstrated efficacy and tolerability, and these promising results led to the development of chemically similar compounds such as lenalidomide and pomalidomide [40].

IMiDs are often used in combination with proteasome inhibitors and/or steroids for the upfront treatment of MM, and as maintenance therapy following autologous stem cell transplants [41,42]. They have demonstrable anti-angiogenic, anti-proliferative, and immunomodulatory effects as they bind to a primary protein target termed cereblon, an E3 ubiquitin ligase complex which leads to the ubiquitination and degradation of Ikaros (IKZF1) and Aiolos (IKZF3), two transcription factors that maintain MM cell function [43]. IMiDs also inhibit MM bone lesions either directly by inhibiting osteoclast maturation or indirectly by reducing MM tumour burden [9]. The concurrent treatment of bisphosphonates with other anti-angiogenic agents such as bevacizumab and tyrosine kinase inhibitors such as sunitiinb and sorafenib has been associated with an increased risk of developing ONJ. The use of lenalidomide has been shown to be independently associated with MRONJ in isolation from other drugs [2,23].

#### 5.1.4. Steroids

The concomitant use of glucocorticoids has been associated with an increased MRONJ risk [1]. This is of particular interest in MM as bone modifying drugs are often administered in association with systemic therapies and dexamethasone, which may potentiate the MRONJ risk in this cohort. Glucocorticoids have both immunosuppressive and anti-inflammatory properties and have been shown to suppress the production of VEGF (vascular endothelial growth factor), directly affecting angiogenesis [44]. High-dose dexamethasone is known to cause bone resorption via the upregulation of the interaction between RANK and RANK-L, and downregulation of OPG (osteoprotegerin), therefore inhibiting osteoblastogenesis and stimulating osteoclastogenesis [9].

### 5.2. Procedures

Dental extractions are the most common initiating event for MRONJ, with other dental procedures such as periodontal debridement and surgery, implant placement, and soft tissue injury from ill-fitting dentures or other prostheses, also implicated [23]. Although MRONJ can occur spontaneously, the risk of spontaneous MRONJ is less likely than following an invasive procedure such as dental extraction [15,25]. A systematic literature review including 3198 cases of bisphosphonate-related ONJ found that 61.7% of cases were related to a prior dental extraction, 14.8% occurred spontaneously, with trauma from protheses (7.4%), history of dental surgery (7.2%), periodontitis (5.0%), and dental implant-related treatment (3.9%) accounting for the remainder of the cases [45].

### 5.3. Comorbidities, Vitamin D Deficiency, Other Medical, Age, Gender

Comorbid diseases including non-insulin dependent diabetes, cardiovascular and respiratory disease, kidney disorders, haematological disorders, tobacco consumption, and mental health illness are reported to significantly increase the odds of developing MRONJ [46]. MRONJ is reported more commonly in older patients, independent of other risk factors [13]. MRONJ is also more common in women than men; however, this is thought to be related to the increased use of BMA in women with osteoporosis and metastatic breast cancer [1].

### 5.4. Oral Health, Inflammation, and Infection

In patients with cancer, pre-existing periodontal or periapical infection was indicated as a risk factor for developing MRONJ in 50% of cases [1], and poor oral health (higher decayed, filled, missing index) has been associated with more advanced cases of MRONJ [47]. Periodontal bone loss and an increased number of missing teeth has been strongly associated with the risk of developing MRONJ in myeloma patients on bone modifying agents [48]. Determining the cause and effect of infection in such cases is challenging as the predisposing dentoalveolar infection initiates the extraction, which is an independent risk for MRONJ. Although inflammation, infection, and the inhibition of angiogenesis all play a role in MRONJ, the order in which these events occur is yet to be elucidated [24].

### 5.5. Anatomical Factors

Certain anatomical sites are more susceptible to MRONJ including the posterior mandible, mylohyoid ridge, and sites of bony prominences including palatal and lingual tori and buccal exostoses as shown in Figure 3a–c [49,50]. These bony prominences are prone to laceration and ulceration as they are covered with only thin oral mucosa that is poorly vascularised. The capacity to respond to trauma is compromised, frequently resulting in inflammation and secondary infection, ischemia, and exposure of the underlying bone [51]. In addition to these sites, the cortical bone thickness has been shown in one study to be a reliable predictor for risk of initiating and progression of MRONJ when comparing MRONJ and non-MRONJ groups [51].

### 5.6. Genetic Factors

Apart from these clinical factors, genetic polymorphisms in farnesyl pyrophosphate synthase or cytochrome P450 CYP2C8 genes have been shown to increase the predisposition in developing MRONJ. However, these studies are limited by a small sample size and further robust data are still needed to predict which individuals will develop MORNJ [52].

## 6. Staging

Since the first staging system was proposed by the American Association of Oral and Maxillofacial Surgeons (AAOMS) in 2009, further revisions have been made to include Stage 0 (prodromal disease) to include those patients at risk of progressing to more advanced MRONJ and an ‘At Risk’ stage which includes all people who have been exposed to oral or IV antiresorptive or anti-angiogenic therapy [1]. Table 2 summarises the clinical and radiographic characteristics of each stage. Figure 4a–e illustrates the clinical and/or radiographic presentation of MRONJ.

## 7. Prevention

Prior to commencing BMA all patients should have a comprehensive dental examination [3]. The goal should be to optimise oral health and eliminate any potential source of infection. The oral and dental examination should include an orthopantomogram and intra-oral radiographs of specific teeth if indicated. Treatment may include:Extraction of teeth with poor restorative or periodontal prognosisExtraction of teeth that are non-functional and pose a future infection riskRemedial dental work to stabilise the dentitionImprove or replace ill-fitting prosthesesInstruction on correct oral hygiene methods including the use of interproximal cleaning and high-fluoride toothpasteDietary counselling on caries risk, reducing sugary drinks and snacksDiscussion about modifiable risk factors including tobacco and alcohol cessation

If invasive procedures are performed, a clinical review of bone and soft tissue healing is essential prior to commencing BMA. Coordinating these procedures with the medical specialist is of the utmost importance.

Patients should see their dental practitioner every six months when education about oral health and dental hygiene should be reinforced. A thorough mucosal examination is necessary to assess for sites of exposed bone or fistulas. Myeloma patients who are being treated with medications associated with MRONJ who develop dental symptoms must be managed promptly. Any routine dental treatment, for example, restorative treatment, may be managed by the dental practitioner; however, invasive treatment or suspected MRONJ, warrants referral to a specialist oral and maxillofacial surgeon. Currently, there is limited evidence for initiating a ‘drug holiday’ prior to performing invasive dental procedures; however, this should be discussed with the haemato-oncologist and decided on in a case-by-case basis [53]. The use of a C-terminal telopeptide test (CTX) prior to dental procedures for patients on bisphosphonates is not predictive of bone healing or for determining MRONJ risk [1,3,54]. It is advisable that any surgical treatment be performed with an atraumatic approach by raising a full-thickness mucoperiosteal flap, removing the tooth, smoothing sharp bone margins, thoroughly irrigating the socket with sterile saline, and achieving primary wound closure [13]. Appropriate post-operative wound care is essential with clinical follow-up to observe healing at appropriate time points after the procedure.

Periodontal surgery, dental implant placement, peri-implantitis, and the removal of implants, are all considered to increase the risk of developing MRONJ in patients on antiresorptive therapy for cancer [13].

## 8. Management of MRONJ

### 8.1. Conservative

Conservative measures are indicated at every stage of the MRONJ treatment timeline. Optimising oral health is critical. Regular toothbrushing and interproximal cleaning with a prescription-strength fluoride toothpaste if indicated, and reducing cariogenic foods and drinks are the most important ways to reduce the risk of dental disease and limit the need for future invasive dental treatment [3]. The use of antimicrobial mouthwash might be indicated and close monitoring of the site at regular time intervals is recommended. Patient education about modifiable risk factors should be conducted including counselling about tobacco and alcohol cessation [3].

### 8.2. Surgical

The indication for surgical procedures should be calculated based on clinical findings and patient symptoms. Sequestrectomy and the removal of associated teeth is indicated if the bone fragment is mobile or causing trauma to the adjacent soft tissues (Figure 3). A recent systematic review and meta-analysis found that when compared with a non-surgical approach to management, surgical treatment improved outcomes, was associated with higher odds of complete resolution, and reduced the likelihood of recurrence [55]. For more advanced stages of MRONJ, surgical debridement or resection with or without reconstruction can encourage resolution [1]. Surgical debridement in association with antimicrobials is also indicated to reduce the total volume of infected bone in some instances to improve the efficacy of antimicrobial therapy [3]. A histological analysis of bone sequestrum is required to eliminate a diagnosis of malignancy or metastatic disease.

### 8.3. Antimicrobial

Although infection is not considered to be the primary cause of osteonecrosis, colonisation of the necrotic bone is common and adjunctive antimicrobial therapy may prove useful in its management [3]. Microbial cultures should be conducted to confirm the sensitivity of microorganisms and allow the initiation of the most appropriate antibiotic therapy. Different types of antibiotics have been recommended including penicillin, metronidazole, clindamycin, doxycycline, erythromycin, and spiramycin [3,55]. One randomised controlled trial demonstrated good compliance and resolution of MRONJ with spiramycin therapy and the authors recommended this line of treatment in cases where previous antimicrobial therapy had failed [56]. Other studies have demonstrated a high prevalence of *Actinomyces* spp. in the necrotic bone of MRONJ patients and support the use of a prolonged antimicrobial regimen of either systemic Augmentin or Clindamycin for four weeks prior to surgical debridement as the mainstay of treatment [57]. The involvement of infectious disease specialists as part of the multidisciplinary team is encouraged for optimal patient outcomes.

## 9. Emerging Therapies for MRONJ

### 9.1. Teriparatide

There have been some promising outcomes for patients treated with teriparatide for stage II-III MRONJ who received oral bisphosphonates for osteoporosis [58]. Teriparatide, a bone anabolic agent, is a synthetic polypeptide hormone containing the 1–34 amino acid fragment of recombinant parathyroid hormone [58]. Teriparatide stimulates osteoblasts and favours the reabsorption of calcium from the kidneys and gut leading to bone formation and re-modelling. Teriparatide eventually leads to increased bone resorption as the processes are inextricably linked. This is due to the ability of the parathyroid hormone to stimulate RANKL by cells of osteoblast lineage [29].

Daily subcutaneous injection had significantly greater improvement in MRONJ outcomes compared with weekly dosing in a small, randomised trial of osteoporotic patients [59]. In a double-blind, randomised-controlled trial, teriparatide was associated with an improved rate of resolution of MRONJ lesions over a 52-week period, with only mild adverse events reported [60]. However, caution regarding teriparatide should be advised as high levels of PTH may be a potential risk factor for MM, as Kang et al. demonstrate that high PTH levels may facilitate the growth of myeloma cells via the secretion of IL-6 and high PTH levels at diagnosis correlated with poorer progression free survival. Further studies of the use of PTH are still needed in MM given its efficacy in patients with osteoporosis [61].

### 9.2. Hyperbaric Oxygen Therapy (HBO)

Hyperbaric oxygen therapy has been trialed in conjunction with surgery and antibiotics in the management of MRONJ and was found to be a useful adjunct in pain relief and healing. Although there was no statistical difference between the control and treatment groups, the size and number of exposed bone sites, quality-of-life, and pain measures were improved in the HBO versus the control groups [62]. A systematic review of HBO found good tolerance for this treatment modality and complete resolution in almost half of the cases; however, the quality of the evidence was low [63]. A Cochrane review concluded there is insufficient evidence to support or refute the benefit of HBO as an adjunct to conventional therapy in the management of MRONJ [64]. As the level of evidence for HBO therapy is low and given the onerous treatment regimen required for this mode of treatment, it is not indicated as a mainstream management strategy at this stage.

### 9.3. Plasma Rich Fibrin (PRF), Photo Biomodulation (Low-Level-Laser Therapy), Antimicrobial Photodynamic Therapy

The use of a plasma rich fibrin during surgical wound closure following extractions or to minimise its recurrence during debridement of an established necrosis has been suggested to prevent MRONJ. To date there is insufficient evidence to support or refute the benefit of this technique [64]. In a recent study, however, it was found that a combination of antibiotic therapy, surgery, PRF, and photo biomodulation led to a statistically significant improvement in outcomes when compared with either antibiotics plus surgery or antibiotics plus photo biomodulation [65]. A systematic review of laser therapy in the management of MRONJ concluded that superior outcomes were achieved when photo biomodulation was used in conjunction with surgical and/or conservative drug therapy compared with surgery alone [66]. The authors suggest that combined treatment with antibiotics, photo biomodulation, and minimally invasive surgery should be the gold-standard of care for early stage MRONJ [66]. Other studies have also supported these findings; however, many are isolated reports or case studies, and evidence is lacking from controlled clinical trials. Antimicrobial photodynamic therapy and PBM protocols were reported as effective methods for both preventing and treating MRONJ lesions in the early stages with no adverse outcomes reported in a prospective study of patients on long-term antiresorptive treatment [67].

### 9.4. Pentoxifylline and Tocopherol

The combination therapy of pentoxifylline and tocopherol has been used with success in the treatment of refractory osteoradionecrosis [68]. Originally approved for use in the management of peripheral artery disease, pentoxifylline inhibits inflammation and decreases fibrosis and improves peripheral blood flow by three key mechanisms: increased vasodilation, reduced blood viscosity, and increased red blood cell flexibility [69]. Tocopherol also reduces inflammation and decreases fibrosis, and is a known potent oxygen scavenger, reducing cell damage from free radicals [69]. There have been some observational studies of small sample sizes reporting success with this mode of treatment with or without adjuvant antimicrobial treatment [69,70,71]. In all patients, there was a decrease in the size of the bone exposed and a reduction in symptoms. A recent systematic literature review concluded that pentoxifylline and tocopherol are potentially useful in the non-surgical management of MRONJ [72].

### 9.5. Mesenchymal Stem Cell Therapy

Mesenchymal stem cells (MSC) are multipotent stem cells used in regenerative therapies and are isolated from bone marrow or adipose tissue but are also present in other tissues [73]. There is an emerging role of the use of mesenchymal stem cells in the treatment of MRONJ as MSC grafts have been shown to be beneficial in recruiting and stimulating local or regional endogenous cells to differentiate into osteoblasts promoting bone formation, bone re-modelling, and have immune-modulatory properties that decrease inflammation [74]. These therapies are still in development with early studies performed in mice and pig models with promising results [75,76].

## 10. Conclusions

MRONJ is common in MM patients with an incidence of between 4.9–20.5%. As MM is the second most common haematological malignancy there are a significant number of patients at riskDuration of exposure to antiresorptive therapy is a major risk factorPrevention is better than a cureMRONJ requires expert management with a multidisciplinary team

## Figures and Tables

**Figure 1 cancers-13-04479-f001:**
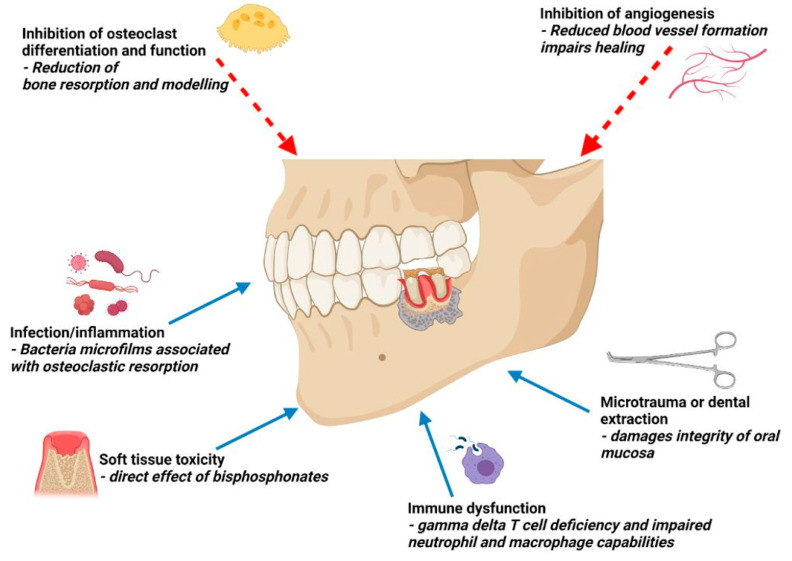
Proposed pathogenesis of MRONJ. (created with BioRender.com, accessed on 6 August 2021).

**Figure 2 cancers-13-04479-f002:**
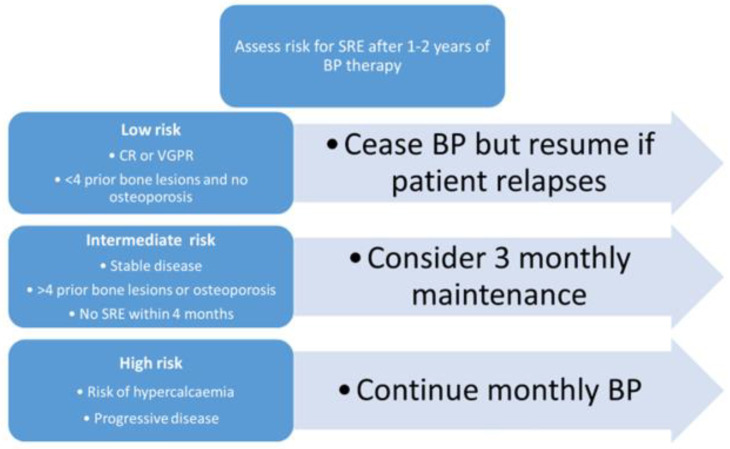
Risk stratification for development of further skeletal related events (adapted from Dickinson et al. [38]).

**Figure 3 cancers-13-04479-f003:**
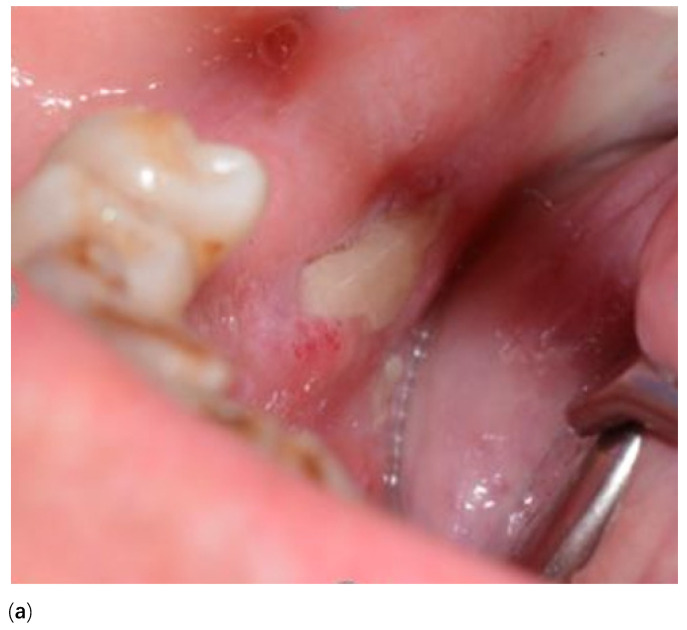
(**a**) MRONJ of the internal oblique ridge right mandible. (**b**) MRONJ of labial bony exostosis. (**c**) MRONJ of the palatal torus (maxilla).

**Figure 4 cancers-13-04479-f004:**
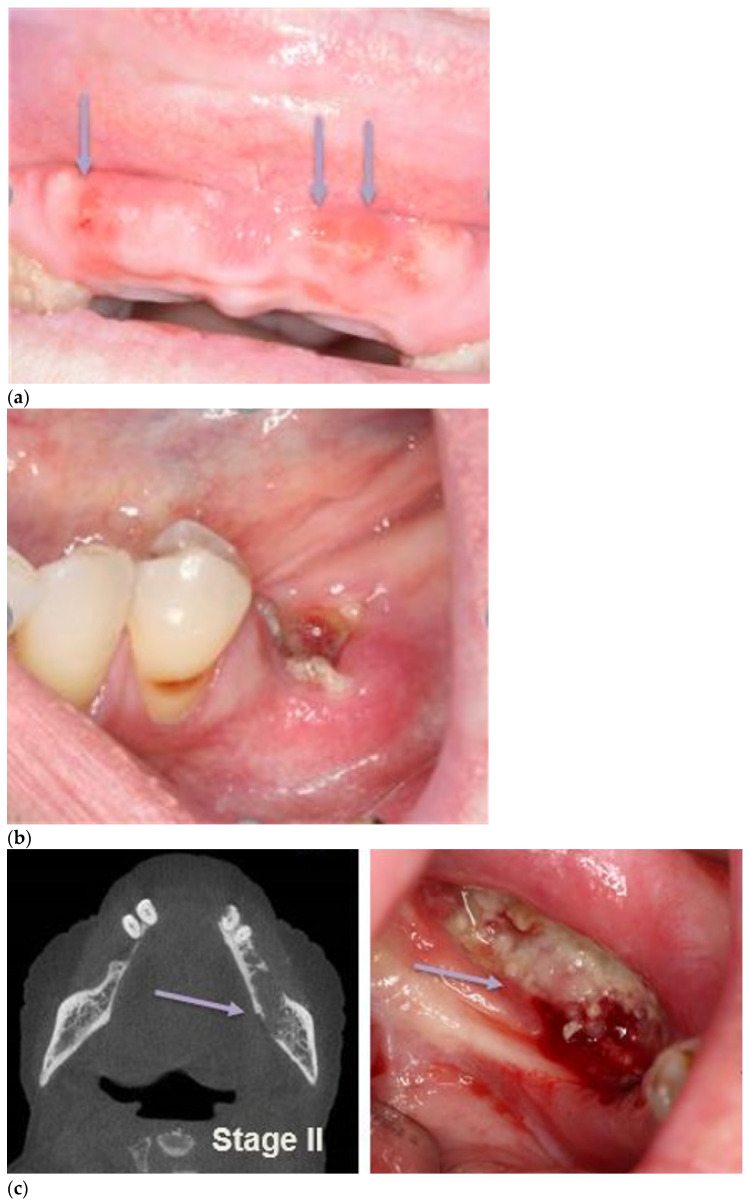
(**a**) Stage I MRONJ presence of fistulas on the maxillary alveolar ridge extending to the underlying bone. (**b**) Stage I MRONJ: site of exposed bone left mandible distal to the premolar that is confined to the alveolar bone superior to the inferior alveolar nerve canal. (**c**) Stage II: MRONJ exposed bone involving the ascending ramus of the left mandible showing both the radiographic and clinical presentation. (**d**) Stage III: cone beam CT demonstrating MRONJ of the left mandible extending below the inferior dental nerve canal. (**e**) Stage III MRONJ; cone Beam CT of the right alveolar processes showing MRONJ associated with upper right molars and the right maxillary sinus.

**Table 1 cancers-13-04479-t001:** Risk of developing MRONJ in patients on bisphosphonate therapy (adapted from Mavrokokki et al. [8]).

Indication for Bisphosphonate Therapy	Risk of Spontaneous ONJ	Risk Following Dental Extraction
All patients	0.05–0.10%	0.37–0.8%
Osteoporosis	0.01–0.14%	0.09–0.34%
Paget’s disease	0.26–1.8%	2.1–13.5%
Malignancy	0.88–1.15%	6.67–9.1%

**Table 2 cancers-13-04479-t002:** MRONJ staging, clinical signs/symptoms, radiographic findings, and management strategies.

Stage	Clinical Signs/Symptoms	Imaging Findings	Management
0	Nil exposed bone Non-specific clinical findings/symptoms Mobile teeth, periapical or periodontal fistula where dental cause is excluded	Non-specific radiographic changes including sclerotic alveolar bone, defined extraction socket, alveolar bone loss in absence of periodontal bone disease, altered trabecular bone appearance, dense bone or absence of bone in extraction sockets, osteosclerosis of alveolar bone, thickening of lamina dura that obscures periodontal ligament space.	Systemic antimicrobial treatment Systemic pain relief Optimise oral/dental health and hygiene
1	Bone exposure/fistula to bone, nil suppuration, asymptomatic	Radiographic findings as stage 0 may be present localised to alveolar bone	Anti-bacterial mouthwash Clinical follow-up every 3 months Optimise oral/dental health and hygiene
2	Bone exposure/necrosis with associated pain, infection, erythema, +/− purulent discharge	Radiographic findings as stage 0 may be present localised to alveolar bone	Anti-bacterial mouthwash Symptomatic treatment with oral antibiotics Pain management Local debridement to relieve soft tissue trauma Optimise oral/dental health and hygiene
3	Exposed and necrotic bone or fistula that probes to bone with associated pain, and infection AND necrotic bone extending beyond the alveolar process to either the inferior border of mandible or ramus, or zygoma of the maxilla resulting in pathological fracture, oral antral or oral nasal communication, or osteolysis extending to the inferior border of the mandible or floor of the maxillary sinus	Osteosclerosis/osteolysis of adjacent bony structures, pathologic fracture, osteolysis extending to maxillary sinus or nasal floor	Anti-bacterial mouthwash Symptomatic treatment with oral antibiotics Pain management Surgical debridement or resection/reconstruction Optimise oral/dental health and hygiene

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
