# Peer review of "(untitled)"

_cancers, 2021, doi:10.3390/cancers13174479_

Round 1

Reviewer 1 Report

 This is a good review Fig 3c is not clear & is probably best deleted The references regarding the use of CTX are weak Best to follow Yarom & Marx 

Author Response

Dear Reviewer,

Thank you for your comments and suggestions. I have removed figure 3c as suggested and referenced the MASCC guidelines and the Ruggerio paper regarding the CTX testing as both these papers are key references for MRONJ.

Kind regards,

The Authors

Reviewer 2 Report

Congratulations. The manuscript is well organized and the clinical and basic knowledge is comprehensively reviewed. This article will help physicians to identify, prevent, and treat the MRONJ induced by oncological therapies for multiple myeloma.  

The author made a great review about MRONJ (Medication-related osteonerosis of the jaw). However, the title of this article is the review the potential side effects induced by Myeloma therapies. The manuscript did not focus on the "ONJ induced by Myeloma therapies" .
There are some problems need to be clarified

1: The mechanism and pathophysiology of ONJ induced by myeloma therapies is the same or different from the other type MRONJ?
2. The prevalence is higher or lower than other type of cancer and cancer therapy as compared with multiple myeloma ?
3. ONJ induced by MM therapies is related to other combined therapy or not ?

otherwise this article can not titled as "Myeloma therapies and their potential for oral and maxillofacial side effects"

As the review and discussion of this manuscript mentioned about MRONJ rather than MRONJ induced by MM therapies.
This is confusing and needs some more discussion.

Author Response

1: The mechanism and pathophysiology of ONJ induced by myeloma therapies
is the same or different from the other type MRONJ?
2. The prevalence is higher or lower than other type of cancer and
cancer therapy as compared with multiple myeloma ?

We have addressed this already under incidence:

MM patients have a higher risk of MRONJ when compared with other patient cohorts taking anti-resorptive medications, with the reported incidence ranging from 4.9-20.5% (4).
3. ONJ induced by MM therapies is related to other combined therapy or not ?

There is not much in the literature specifically about the combined therapies used for myeloma and ONJ risk. We have added some information about combined therapy pertaining to cancer treatment in general and the increased risk.

Recent data suggest that combination therapies might also increase the risk of developing MRONJ, and cause more advanced necrosis especially in the maxilla, however this study was not limited to patients with myeloma (22). Another study of 459 MRONJ cases reported that out of 52 patients undergoing treatment with BMA, 11 had also received lenalidomide, 12 received thalidomide, 11 bevacizumab, 9 everolimus and 9 sunitinib as part of the drug therapy (23).  Although bisphosphonates are the drugs most frequently associated with MRONJ, there is a growing range of non-antiresorptive medications implicated in MRONJ development (24). 

We welcome any suggestions for a more appropriate title for the manuscript if the reviewer has an appropriate alternative. 

Thank you again for the opportunity to submit this paper for review.

Reviewer 3 Report

Medication-related osteonecrosis of Jaw represents an important complication and limitation to the use of antiresorptive therapy in multiple myeloma.

The authors elegantly present the problem, its incidence, pathogenesis as well the risk factors of the medications including bisphosphonates,  monoclonal antibodies and immunomodulators and steroids. They also highlight the different procedures and other factors that pave the way for pathogenesis to develop.

The article, thereafter describes the staging, the prevention and the management strategies to abord the problem including the emerging therapies like Teriparatide, hyperbaric oxygen and PRF and pentoxifylline as well as stem cell therapy.
The article is very well written and represents an important addition in the field.

Author Response

The authors thank you for reviewing this manuscript and providing the positive feedback and comments.